# Phosphorylcholine Monoclonal Antibody Therapy Decreases Intraplaque Angiogenesis and Intraplaque Hemorrhage in Murine Vein Grafts

**DOI:** 10.3390/ijms232113662

**Published:** 2022-11-07

**Authors:** Fabiana Baganha, Thijs J. Sluiter, Rob C. M. de Jong, Louise A. van Alst, Hendrika A. B. Peters, J. Wouter Jukema, Mirela Delibegovic, Knut Pettersson, Paul H. A. Quax, Margreet R. de Vries

**Affiliations:** 1Department of Surgery, Leiden University Medical Center, 2333 ZA Leiden, The Netherlands; 2Einthoven Laboratory for Experimental Vascular Medicine Leiden University Medical Center, 2333 ZA Leiden, The Netherlands; 3Department of Cardiology, Leiden University Medical Center, 2333 ZA Leiden, The Netherlands; 4Aberdeen Cardiovascular and Diabetes Centre, Institute of Medical Sciences, University of Aberdeen, Aberdeen AB25 2ZD, UK; 5Athera Biotechnologies, Bioventurehub, SE-431 83 Mölndal, Sweden

**Keywords:** atherosclerosis, angiogenesis, hemorrhage, CD163+ macrophages, vein graft

## Abstract

Phosphorylcholine (PC) is one of the main oxLDL epitopes playing a central role in atherosclerosis, due to its atherogenic and proinflammatory effects. PC can be cleared by natural IgM antibodies and low levels of these antibodies have been associated with human vein graft (VG) failure. Although PC antibodies are recognized for their anti-inflammatory properties, their effect on intraplaque angiogenesis (IPA) and intraplaque hemorrhage (IPH)—interdependent processes contributing to plaque rupture—are unknown. We hypothesized that new IgG phosphorylcholine antibodies (PC-mAb) could decrease vulnerable lesions in murine VGs.Therefore, hypercholesterolemic male ApoE3*Leiden mice received a (donor) caval vein interposition in the carotid artery and weekly IP injections of (5 mg/kg) PCmAb (n = 11) or vehicle (n = 12) until sacrifice at day 28. We found that PCmAb significantly decreased vein graft media (13%), intima lesion (25%), and increased lumen with 32% compared to controls. PCmAb increased collagen content (18%) and decreased macrophages presence (31%). PCmAb resulted in 23% decreased CD163+ macrophages content in vein grafts whereas CD163 expression was decreased in Hb:Hp macrophages. PCmAb significantly lowered neovessel density (34%), EC proliferation and migration with/out oxLDL stimulation. Moreover, PCmAb enhanced intraplaque angiogenic vessels maturation by increasing neovessel pericyte coverage in vivo (31%). Together, this resulted in a 62% decrease in IPH. PCmAb effectively inhibits murine atherosclerotic lesion formation in vein grafts by reducing IPA and IPH via decreased neovessel density and macrophages influx and increased neovessel maturation. PC-mAb therefore holds promise as a new therapeutic approach to prevent vein graft disease.

## 1. Introduction

Vein grafts are frequently used in central and peripheral bypass surgery, and failure of the grafts remains a problem [1]. Inflammation, intraplaque angiogenesis (IPA) and intraplaque hemorrhage (IPH) are interdependent processes contributing to the development and rupture of atherosclerotic plaques as well as neointimal hyperplasia [2,3,4]. Reduced oxygen availability or hypoxia in the plaque is a direct effect of increased size and active inflammation in the lesions. Triggered by hypoxia, endothelial cells (EC) proliferate and migrate from the adventitia and form neovessels that grow into the lesion to overcome the oxygen demand [2]. However, these neovessels are frequently immature and highly susceptible to leakage, and are therefore the main source of IPH [5]. IPH, defined as the extravasation of blood, is a source of hemoglobin (Hb) and other erythrocyte membrane components, such as free cholesterol and phospholipids to the plaques [6,7]. Intake of Hb by macrophages drives upregulation of the CD163 scavenger receptor and leads to a distinct macrophage phenotype, M(CD163+) [8]. These macrophages produce and secrete high levels of VEGFA via the HIF1a pathway, thereby promoting a vicious cycle of angiogenesis, IPH and inflammation [8]. Moreover, since Hb is a strong oxidizer, due to its high iron content, it increases the presence of oxidized phospholipids [9,10].

During oxidation of phosphatidylcholine (the most abundant phospholipid in oxLDL [11] and cell membranes [12]), phosphorylcholine (PC) headgroups are exposed. These PC epitopes, recognized as DAMPs [13], trigger complex immunoinflammatory responses, induce toxic oxidative stress, apoptosis, EC activation [14] and dysfunction [15]. Moreover, these PC epitopes also mediate the oxLDL uptake by macrophage scavenger receptors. PC contributes via all these processes to native and accelerated atherosclerosis, the latter can occur after an intervention, such as stenting or bypass surgery [1]. Naïve atherosclerosis develops over decades, whereas accelerated atherosclerosis in vein grafts or in stents can be observed within months to years [16]. Apart from several discrepancies, the process of both native and accelerated atherosclerosis is relatively similar. Atherosclerosis is characterized by endothelial dysfunction and oxLDL-presence. Endothelial dysfunction leads to infiltration of leukocytes, predominantly macrophages, that aim to clear the oxLDL in the subendothelial layer. Upon oxLDL recognition and internalization, macrophages undergo metabolic and functional reprogramming, ultimately resulting in foam cell formation [17]. In addition, oxLDL can result in mitochondrial DNA damage in macrophages leading to cell death, demonstrating the crucial role of phospholipids in atherogenesis [18].

PC epitopes can be cleared by natural IgM antibodies against PC, produced and released by B1 cells [19,20]. These natural antibodies have been shown to control oxidative stress and inhibit macrophage oxLDL uptake, thereby preventing foam cell formation and enhancing efferocytosis [21,22,23,24]. In ApoE^-/-^ mice, immunization with anti-PC IgM reduced vein graft (VG) size and plaque inflammation [24]. Moreover, low levels of IgM anti-PC were associated with VG failure in a large human cohort [25].

To construct a therapeutic anti-PC we developed an IgG human monoclonal antibody against PC (PC-mAb) which specifically targets oxidized phospholipids and PC and decreases inflammation and accelerated atherosclerosis in mice [26]. PCmAb shows safe pharmacokinetics in rats and cynomolgus monkeys [26], and is currently in clinical development (EudraCT Number:2018-003676-12; ClinicalTrials.gov Identifier:NCT03991143). This therapeutic IgG anti-PC was also shown to decrease inflammation in naïve atherosclerosis in mice [27]. Therefore, we hypothesized that PC-mAb might modulate and normalize IPA and IPH, as anti-PC IgM modulates the progression of vascular inflammation, stabilizing atherosclerotic lesions.

We previously demonstrated that murine hypercholesterolemic ApoE3*Leiden VG lesions highly resemble the human atherosclerotic unstable plaques with pathological intimal thickening, severe inflammation, leaky neovessels and hemorrhage [28,29,30]. Here, we studied the role of PC-mAb on inflammation, IPA, and IPH in ApoE3*Leiden VG lesions. Moreover, we investigated the isolated effect of PC-mAb in in vitro angiogenesis assays and CD163+ macrophage cultures.

## 2. Results

### 2.1. PC-mAb Decreases Vein Graft Thickening and Increases Lumen Area

PC-mAb treatment did not affect bodyweight or cholesterol levels (Appendix A). In both groups (CTRL: n = 3, PC-mAb: n = 4) mice were excluded from further analysis due to fully occluded vein grafts as a result of thrombosis. PC-mAb effects on vein graft morphometry and vessel wall remodeling were assessed using the Masson’s Trichrome staining (Figure 1A).

Passive immunization with PC-mAb did not affect Vessel Area and thus negative remodeling (Figure 1B) but decreased the Vessel Wall Area by 25% (*p* = 0.0246, Figure 1C), which resulted in a 22% decrease in Intimal Hyperplasia (*p* = 0.0398) in comparison to the CTRL group (Figure 1D). Moreover, the Lumen Area was significantly increased (32%) by PC-mAb treatment in comparison with the CTRL group (*p* = 0.0236, Figure 1E).

### 2.2. PC-mAb Increases Collagen Content

To assess PC-mAb effects on the lesion composition, we analyzed stable and unstable plaque features [28,31], such as % ACTA2 area (Figure 2) as well as a semi-quantitative analysis of specific advanced atherosclerotic lesion features such as foam cells, fibrin, calcification (Appendix A). Although the % ACTA2 area—consisting of cells that are the main producers of collagen—did not differ between the two groups, the % Collagen was increased by PC-mAb treatment (by 18%) in comparison to the CTRL group (*p* = 0.0404, Figure 2A–C). The % Collagen in the intimal hyperplasia as well as collagen maturation (assessed by polarized light) was similar in both groups (Appendix A).

### 2.3. PC-mAb Decreases ICAM-1 and VCAM-1 Expression

To investigate PC-mAb effects on inflammation, we determined the relative presence of macrophages in the vessel wall (% MAC-3). PC-mAb treatment significantly decreased (by 31%) the % MAC-3 in comparison to the CTRL group (*p* = 0.0333, Figure 3A,B).

Next, we quantified the expression of adhesion molecules ICAM-1, VCAM-1 and MCP-1, the most potent chemoattracting chemokine for monocytes in the vein graft lesions. PC-mAb treatment reduced ICAM-1 (29%, *p* = 0.0104) and VCAM-1 (36%, *p* = 0.0073) expression in comparison with the CTRL group (Figure 3C,D,F). However, the % MCP-1 did not vary between the two groups (Figure 3E,F).

### 2.4. PC-mAb Decreases Intraplaque Angiogenesis and Intraplaque Hemorrhage

To evaluate PC-mAb effects on IPA, we measured the density of the neovessels in the vein grafts expressed as % Neovessels as well as the neovessels that lacked pericyte coverage (% Immature Neovessels) and we evaluated Intraplaque Hemorrhage in the VG lesions by scoring extravasated erythrocytes in a triple staining (Figure 4A).

Quantification of the % Neovessels revealed a 34% decrease in the PC-mAb group in comparison with the CTRL group (*p* = 0.0006, Figure 4B). Quantification of % Immature vessels showed a 31% decrease in PC-mAb-treated group in comparison with the CTRL group (*p* = 0.0042, Figure 4C). PC-mAb treatment decreased Intraplaque Hemorrhage presence and severity as scored in Figure 4D. In the CTRL group, 92% (11/12) of mice presented extravasated erythrocytes, compared with 55% (6/11) in the PC-mAb group. Moreover, in the CTRL group, Intraplaque Hemorrhage severity was scored as ≤1 in 54% (6/11) of the mice, as ≤2 in 36% (4/11) and as ≤3 in one mouse. In the PC-mAb group, Intraplaque Hemorrhage severity was scored as ≤1 in 83% (5/6) of mice presented and as ≤2 in one mouse.

### 2.5. PC-mAb Decreases EC Metabolic Activity and Migration and Sprouts Formation

Since the observed changes in plaque size and inflammation can reduce oxygen demand and consequently decrease IPA, we studied the PC-mAb effects on EC behavior and neovessel sprouting in vitro.

HUVEC metabolic activity in the MTT assay was reduced by 22% by 10 µg/mL of PC-mAb (*p* = 0.0875) and by 28% by 100 µg/mL of PC-mAb (*p* = 0.0415), as shown in Figure 5A. HUVEC migration—or the ability of wound closure—was decreased by, resp., 54% and 33% when treated with 10 µg/mL (*p* = 0.0395) and 100 µg/mL of PC-mAb (*p* = 0.155). OxLDL treatment of HUVEC results in an enhancement of migration of 52% when compared to CTRL conditions. Treatment with PC-mAb decreased oxLDL stimulated HUVEC migration dose-dependently by 31% (10 µg/mL, *p* = 0.0439) and 73% (100 µg/mL, *p* = 0.0001) when compared to the oxLDL-CTRL group (Figure 5B). In the aortic ring assay (Figure 5C), the number of sprouts was dose-dependently decreased by PC-mAb in comparison to the CTRL group, with a 50% decrease for the 100 µg/mL treated segments.

### 2.6. PC-mAb Targets M(CD163) Macrophages In Vivo and In Vitro by Decreasing CD163 Expression

It has been shown that CD163+ macrophages not only promote leucocyte infiltration but also induce angiogenesis and vessel permeability by secreting VEGFA [8]. Therefore, we determined the %M(CD163) macrophages in the VG lesions. %M(CD163) was reduced by 23% in the PC-mAb group in comparison to the CTRL group (*p* = 0.0014, Figure 6A,B).

Since PC-mAb decreased the amount of extravasated erythrocytes in VG lesion (Figure 4D), we analyzed the in vitro effects of PC-mAb on the upregulation of CD163 receptor in macrophages. Unstimulated THP1 cells already express CD163 receptor and treatment with PC-mAb did not change CD163 expression (Figure 6C,D). Stimulation with HH-enriched media increased CD163 expression by 86% (*p* = 0.0226) in comparison to the CTRL group. Under this condition treatment with PC-mAb decreased CD163 expression by 50% (10 µg/mL, *p* = 0.0130) and 44% (100 µg/mL, *p* = 0.0314) in comparison with the HH group. M(CD163+) are active secretors of VEGFA, therefore we quantified VEGFA protein levels in the macrophage medium. Stimulation with a HH-enriched medium shows an increase in VEGFA levels by 68% in comparison to the CTRL group. Treatment with both PC-mAb concentrations decreased VEGFA levels in comparison to the HH group (15% and 20%, respectively for 10 and 100 ug/mL PC-mAb).

## 3. Discussion

In the present study, we provide evidence that in the hypercholesterolemia ApoE3*Leiden vein graft model, PC-mAb has strong effects (1) on plaque stability and inflammation by reducing macrophage content, (2) on IPA by decreasing the neovessel density and by improving their maturity and (3) on IPH by decreasing erythrocytes extravasation. Further, we demonstrate for the first time that PC antibodies can reduce the activity of scavenger receptor CD163.

We evaluated PC-mAb effects on VG morphometry and morphology and observed that vessel wall thickening (intima + media) was decreased in a beneficial way upon treatment. Interestingly in a human cohort study, the natural PC IgM levels inversely correlated with the presence of intima media thickening [32]. Moreover, the size of the intima layer also changed when treated with PC-mAb, displaying less intimal hyperplasia. Lumen area was increased in the PC-mAb treated group, which is clinically the most relevant parameter since this directly improves blood flow. Loss of vein graft patency is an important clinical problem for which there is no effective treatment. Low levels of the IgM anti-PC are associated with an increased risk of loss of VG patency [25].

PC-mAb treated plaques display significantly increased levels of collagen and decreased MAC-3 content. These findings are in accordance to previous studies that reported that anti-PC binds to PC epitopes on oxidized phospholipids, inhibiting inflammatory signaling [19] and blocking uptake of oxLDL and foam cell formation [19,24,33]. The IgG PC-mAb is also able to block oxLDL uptake by macrophages unlike the T15/E06 based IgG and binds late apoptotic cells, but not normal or ‘early’ apoptotic cells [3]. We recently showed that PC-mAb prevented macrophage influx after myocardial infarction via reduced systemic inflammatory responses [34,35].

Oxidized phospholipids are known to trigger ECs to undergo inflammatory activation [36]. Therefore, we assessed the expression of adhesion molecules such as ICAM-1 and VCAM-1, and the chemokine MCP-1, which represent important triggers to attract monocytes in early (expressed on the lumen surface and in VSMCs) and late lesion development (expressed on neovessels endothelium). We showed that PC-mAb treatment decreases VCAM-1 and ICAM-1 levels in the vessel wall, which can be a direct cause for the lower macrophage content in the plaque. Moreover, since there is a direct relationship between entry of monocytes/macrophages and leakage of neovessels [7,37], a reduction in the amount of macrophages may be a direct result of reduced IPA in the plaque by PC-mAb.

Furthermore, PC-mAb decreases the presence of M(CD163+) in VG lesions significantly. When we tested the effect of PC-mAb on cultured macrophages stimulated with Hb:Hp complexes, PC-mAb significantly decreased CD163 expression. This suggests that PC epitopes may also be involved in the CD163 scavenger activity, as it happens for other scavenger receptors, such as CD36 [38,39]. According to Guo et al., CD163+ macrophages perpetuate IPA and IPH through the secretion of VEGFA [8]. In our in vitro setup, stimulation with Hb:Hp complexes increased macrophage VEGFA secretion levels which could be prevented by treatment with PC-mAb, which thus both reduces CD163 expression and its activity.

Since it is well known that directly after vein graft procedure the endothelium is damaged and activated, we investigated the effects of PC-mAb on human venous endothelial cells [1]. The anti-angiogenic capacity of PC-mAb was proven by a decrease in EC proliferation (or metabolic activity in the MTT assay), EC migration and neovessel sprouting. Moreover, oxLDL induced HUVEC migration was reduced in presence of PC-mAb demonstrating that PC-mAb, by neutralizing oxLDL-PC epitopes, reduces EC angiogenic behavior. It has been shown by Schnitzler et al. that PC containing oxidized phospholipids increase EC metabolism by inducing expression of glycolytic enzymes which can be prevented by anti-PC [40]. Immunization with PC-mAb may thus decrease cell metabolism and, therefore, oxygen demand in the plaque. IPA in the vein grafts is induced by hypoxia as shown by pimonidazole expression [29]. We observed a clear reduction in neovessel density upon PC-mAb. These neovessels are leaky [41] and can be therapeutically targeted by interference in pericyte coverage [29,30]. Interestingly, PC-mAb has a positive effect on IPH by improved vessel maturation which decreased leakage of erythrocytes in the plaques.

Taken together, our findings reveal that PC-mAb improves plaque stability by decreasing lesion size, inflammation, IPA and IPH, as a result of increased vessel maturation and decreased M(CD163) presence. In conclusion, our findings explain the beneficial PC-mAb effects on advanced atherosclerotic lesions and specifically as a treatment to improve vein graft patency.

## 4. Materials and Methods

### 4.1. Animals

All animal experiments were performed in compliance with Dutch government guidelines and the Directive 2010/63/EU of the European Parliament. The institutional committee of the Leiden University Medical Centre approved all the animal experiments licensed under project number (13166). Male ApoE3*Leiden mice (bred in our own colony), 10–16 weeks old, were fed with a diet containing 1% cholesterol and 0.05% cholate (AB diets) to induce hypercholesterolemia for three weeks prior to surgery until sacrifice. All animals received food and water ad libitum. Mice were randomized based on their plasma cholesterol levels (Roche Diagnostics, kit 1489437) and body weight (Appendix A).

### 4.2. Vein Graft Surgery

After three weeks on diet, the mice underwent vein graft surgery, by means of a donor caval vein interposition in the carotid artery of recipient mice, as described before [42]. Mice were anesthetized intraperitoneally with 5 mg/kg of midazolam (Roche Diagnostics), 0.5 mg/kg of dexmedetomidine (Orion Corporation) and 0.05 mg/kg of fentanyl (Janssen Pharmaceutical). After surgery, the anesthesia was antagonized with 2.5 mg/kg of atipamezol (Orion Corporation) and 0.5 mg/kg of flumazenil (0.5 mg/kg, Fresenius Kabi). An amount of 0.1 mg/kg of buprenorphine (MSD Animal Health) was given for pain relieve. Animals were sacrificed 28 days after the surgery, via exsanguination after anesthesia (described above) followed by 3 min of in vivo perfusion-fixation with PBS and 4% formaldehyde (100496, Sigma-Aldrich, Saint Louis, MO, USA). The vein grafts were harvested and fixed in 4% formaldehyde.

### 4.3. Treatment

Mice were treated with intraperitoneal injections of a human IgG1 phosphorylcholine monoclonal antibody (5 mg/kg ATH3G10, Athera Biotechnologies, Mölndal, Sweden n = 15) at days 7,14 and 21 [26]. As a negative control, sterile 0.9% NaCl (Fresenius Kabi, Bad Homburg vor der Höhe, Germany, n = 15) was used. Based on previous experiments this treatment was comparable to IgG1 control antibodies in an accelerated atherosclerosis model [26].

### 4.4. Histology and Immunostainings in Vein Grafts

Vein graft samples were embedded in paraffin and sequential cross sections (5 µm thick) were taken from the entire length of vein graft. Sectioning was standardized between animals by collecting sections that only contained the venous part of the vein graft was, i.e., starting with the section where no cuff was visible anymore. 12 Ribbons with 20 sections (of 5 µm, a total length of 100 µm) were sectioned. The sections were distributed over 20 slides in a standardized manner of first section of ribbon 1 on the first slide, section 2 of ribbon 1 on the second slide and so on, resulting in 20 slides containing 12 sections throughout the vein graft. Per staining for each mouse 1 slide was used and the 6 middle sections of the vein graft were used for analysis.

To assess vessel morphometry, vein graft cross-sections were stained with Masson Trichrome (Hematoxylin, Biebrich Scarlet-Acid Fuchsine and Aniline Blue). Using Qwin software (Leica), the following parameters were analyzed: area within the border of the adventitia (Vessel Area), area within the media (Media Area), and area within the luminal border (Lumen Area). From this, Vessel Wall Area (media +intimal hyperplasia) is calculated by subtraction of the Lumen Area from the Vessel Area and Intimal Hyperplasia is calculated by subtraction of the Lumen Area from the Media area. Since the lumen is affected by vessel enlargement, Lumen Area is expressed as a percentage of Vessel Area.

To assess vessel morphology: the presence of Dissections, Fibrin, Foam cells, Chondrocytes and Calcification were semi-quantitively scored in the Masson Trichrome stained sections. No presence was scored as 0, low as 1, intermediate as 2, high as 3. The relative amount of collagen, expressed as a percentage of the Vessel Wall Area and Intimal Hyperplasia (% Collagen) was quantified in Sirius Red stained sections. The maturation of collagen fiber was assessed by polarized light, as previously described28, and normalized to % Collagen. To further specify vessel histology: the relative content of vascular smooth muscle cells (% ACTA2 area) and macrophages (% MAC-3) were analyzed by immunohistochemistry for ACTA2 (1A4, Dako) and MAC-3 (553322, BD Pharmingen), respectively. Additionally, the relative expression of adhesion molecules, such as VCAM-1 (ab134047, Abcam), ICAM-1 (ab25375, Abcam), and monocyte chemokines, MCP-1 (sc-1784, Santa Cruz) was assessed by immunohistochemistry. Immuno-positive areas were quantified with ImageJ software, and normalized for Vessel Wall Area.

To assess IPA and IPH, a triple immunofluorescence staining was used, including CD31 (sc-1506-r, Santa Cruz) to detect neovessels, ACTA2 to evaluate vessel maturation (smooth muscle cell coverage), and the erythrocyte marker TER119 (116202, Biolegend). CD31+ neovessels were manually counted per cross section (six cross sections/vein graft) and expressed as number of neovessel of the corresponding vessel wall area (% Neovessels). Neovessel CD31 + ACTA2- coverage was quantified by manually counting the number of neovessels that were not encompassed by ACTA2 and expressed as number of neovessels of the corresponding vessel wall area (% Immature Neovessels). Intraplaque hemorrhage was regionally assessed using a scoring system accounting for the presence and the number of erythrocytes outside the neovessels. No presence was scored as 0, low number of erythrocytes outside the neovessels (1–10 neovessels with extravascular erythrocytes) was score as 1, intermediate number (11–30 neovessels with extravascular erythrocytes) as 2, high number (>30 neovessels with extravascular erythrocytes) as 3.

For each antibody, isotype-matched antibodies were used as negative controls. Pictures were acquired with the Pannoramic SCAN II (3DHistech).

### 4.5. Cell Culture

For the isolation of human umbilical vein endothelial cells (HUVEC), anonymous umbilical cords were obtained in accordance with guidelines set out by the ‘Code for Proper Secondary Use of Human Tissue’ of the Dutch Federation of Biomedical Scientific Societies (Federa) and conform to the principles outlined in the Declaration of Helsinki. HUVEC were isolated and cultured as described by Welten et al. [43]. In brief, the vein in the umbilical cords was flushed with warm PBS and incubated with 0.75 mg/mL collagenase type II (LS004177, Worthington Biochemical Corporation) for 20 min at 37 °C. Detached EC were washed out of the vessel and left to grow in complete medium [EBM-2 medium (00190860, Lonza) supplemented with EGM BulletKit (CC-3124, Lonza) and 2% of FBS (10082139, ThermoFisher Scientific)] at 37 °C in a 5% CO_2_ humidified incubator. Culture medium was refreshed every 2–3 days. Cells were passed using trypsin-EDTA (T4049, Sigma-Aldrich) at 90–100% confluency. HUVEC were used up to passage three for proliferation and migration assays, and up to passage seven for Western blot.

THP1 cells (88081201, Merck) were seeded at a density of 106 cells per ml in 6-well tissue culture plates and incubated with 100 nM of phorbol-12-myristate-13-acetate (PMA, Sigma-Aldrich) for 24 h in complete medium (RPMI with 10% Fetal Calf Serum) for differentiation.

### 4.6. MTT Assay

Cell metabolic activity as a marker for cell proliferation was measured by the reduction of (3-(4,5-dimethylthiazol-2-yl)-2,5-diphenyltetrazolium bromide (M5655, Sigma-Aldrich). HUVEC were seeded in 96-wells plate in complete medium and grown until 80% confluency. To cause cell cycle arrest, cells were incubated for 24 h in EBM-2 medium supplemented with 0.2% FBS. PC-mAb was added in concentrations of 10 µg/mL and 100 µg/mL. After 18 h, cells were incubated with MTT for 4 h. A supernatant fraction was replaced by 0.01 N HCL-isopropanol (258148 and 563935, Sigma-Aldrich) and absorbance was measured at 570 nm by Cytation™ 5 Cell Imaging Multi-Mode Reader (BioTek Instruments).

### 4.7. Migration Assay

For migration assays, HUVEC were seeded in 12-wells plate in complete medium and grown until 100% confluence [44]. To cause cell cycle arrest, cells were incubated in EBM-2 medium supplemented with 0.2% FBS and 24 h later, a scratch-wound was made. PC-mAb was added in concentrations of 10 µg/mL and 100 µg/mL. In the migration assay, HUVEC were stimulated with 5 µg/mL of oxLDL (L34357, ThermoFisher Scientific) to mimic PC presence. Three locations along the scratch-wound were marked per well and scratch-wound closure at these sites were imaged at 0 h and 16 h by using Axiovert 40c Inverted & Phase Contrast Microscope (451207, Carl Zeiss). Average scratch-wound closure after 16 h was calculated by measuring cell coverage at 16 h vs. 0 h using ImageJ.

### 4.8. Aortic Ring Sprouting Assay

The aortic ring assay was performed as described previously [43]. Three ApoE3*Leiden mice, 4–8 weeks old, were anesthetized and the aorta was dissected. Each aorta was cut in 1 mm rings, and serum-starved in Gibco™ Opti-MEM™ GlutaMAX (51985034, ThermoFisherScientific) overnight at 37 °C and 5% CO_2_. On the next day, each ring was mounted in a well of a 96-well plate in 70 µL of 1.0 mg/mL acid-solubilized collagen type-I (11179179001, Roche Diagnostics) in DMEM (12634010, ThermoFisher Scientific). After collagen polymerization, Gibco™ Opti-MEM™ GlutaMAX supplemented with 2.5% FCS and 30 ng/mL VEGF (recombinant mouse VEGF-164) was added with PC-mAb (10 µg/mL and 100 µg/mL). The rings were cultured for 7 days and photographed using Axiovert 40c microscope. The number of sprouts were counted manually. For immunohistochemistry, rings were formalin-fixed and permeabilized with 0.2% Triton X-100 (11332481001, Merck). Rings were stained with ACTA2, CD31 and VE-Cadherin (AF1002, R&D Systems). Extended focus pictures were made with the Pannoramic SCAN II and quantified with Image J.

### 4.9. Protein Expression Analysis

Differentiated THP-1 cells were incubated in HH enriched media (0.1 mg/mL of Hb:Hp (H0267 and SRP6507, Sigma Aldrich) in complete medium) over 6 days. At day 7, 10 µg/mL and 100 µg/mL of PC-mAb was added to the medium and incubated overnight.

Cells were scraped and homogenized in modified RIPA buffer containing sodium-orthovanadate and protease inhibitors. Proteins were separated by SDS-PAGE (4–15%) and transferred to nitrocellulose. Blots were incubated with antibodies against CD163 (93498, Cell Signaling). A peroxidase conjugated secondary antibody was used (31462, 31400, ThermoFisher Scientific). Proteins of interest were imaged with SuperSignal™ West Pico PLUS Chemiluminescent Substrate (34580, ThermoFisher Scientific) and the ChemiDoc™ Touch Imaging using System (1708370, Bio-Rad Laboratories). β-actin (ab8220, Abcam) was used as internal control and blots were quantified with Image J.

Cell supernatant was also collected and VEGFA was measured by a sandwich ELISA (DYC5079-2, R&D Systems, Minneapolis, MN, USA) according to the manufacturer’s instructions.

### 4.10. Statistical Analysis

All data are presented as mean ± standard error of the mean (SEM). Normality was examined using the Shapiro–Wilk normality test. Overall comparisons between groups were performed using unpaired t-test, 1-way ANOVA or 2-way ANOVA on parametric data and a Kruskal–Wallis test for nonparametric data using the statistics software Prism 8.02 (GraphPhad Software Inc., San Diego, CA, USA). *p*-values less than 0.05 were regarded as statistically significant.

## 5. Conclusions

Phosphorylcholine is a pro-inflammatory epitope that acts as a damage associated molecular pattern triggering complex inflammatory responses including macrophage scavenging and endothelial dysfunction. In this study, we investigated the modulatory effects of a fully human IgG1 monoclonal antibody directed against phosphorylcholine (PC-mAb) on VGD. We demonstrate that PC-mAb attenuates IPA—both in vivo and in vitro—and IPH by improving neovessel maturation. Additionally, PC-mAb improves plaque stability via reduced macrophage content, including angiogenesis associated CD163+ macrophage, reduced ICAM and VCAM expression and increased collagen content. PC-mAb therapy may therefore be a valid therapeutic approach to prevent vein graft failure.

## 6. Patents

PCmAB, Athera Biotechnologies AB, patent 9803028.

## Figures and Tables

**Figure 1 ijms-23-13662-f001:**
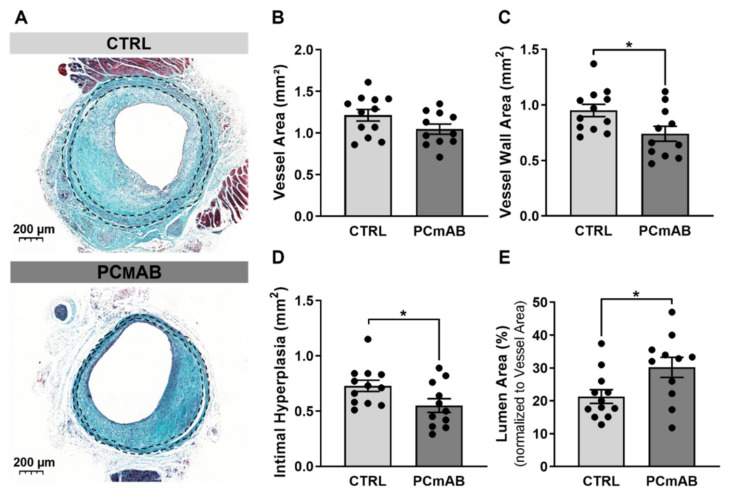
PC-mAb decreases intimal hyperplasia and increases lumen area in vein graft atherosclerosis. Masson Trichrome staining representative vein grafts cross-sections (**A**) of hypercholesteraemic ApoE3*L mice treated with 0.9% NaCl sterile solution (n = 12) and 5 mg/kg of PC-mAb (n = 11). Quantitative measurements of Vessel Area (= Vessel wall area + lumen) (**B**), Vessel Wall Area (**C**), Intimal Hyperplasia (**D**) and Lumen Area (**E**). Data presented as mean ± SEM. * *p* ≤ 0.05 by *t*-test.

**Figure 2 ijms-23-13662-f002:**
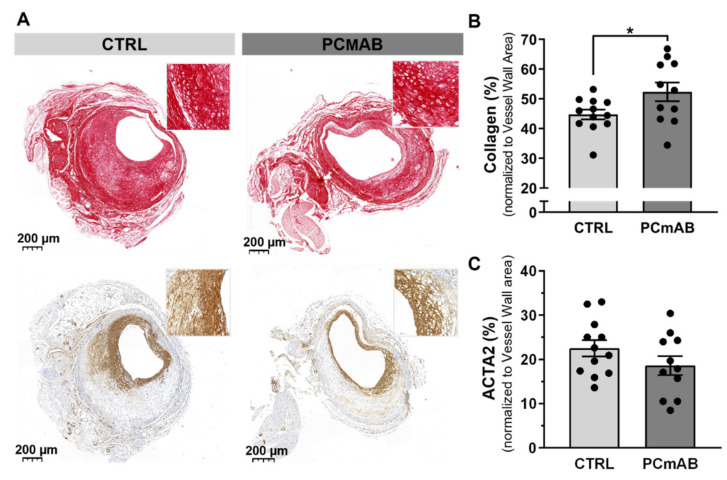
PC-mAb improves plaque stability by increasing collagen content in vein graft atherosclerosis. Representative vein grafts cross sections (**A**) of Sirus Red Staining, Actin alpha 2 (ACTA2) and Masson Trichrome of CTRL (n = 12) and PC-mAb group (n = 11). Quantitative measurements of % Collagen (**B**), % ACTA2 area (**C**). Data presented as mean ± SEM. * *p* ≤ 0.05 by *t*-test.

**Figure 3 ijms-23-13662-f003:**
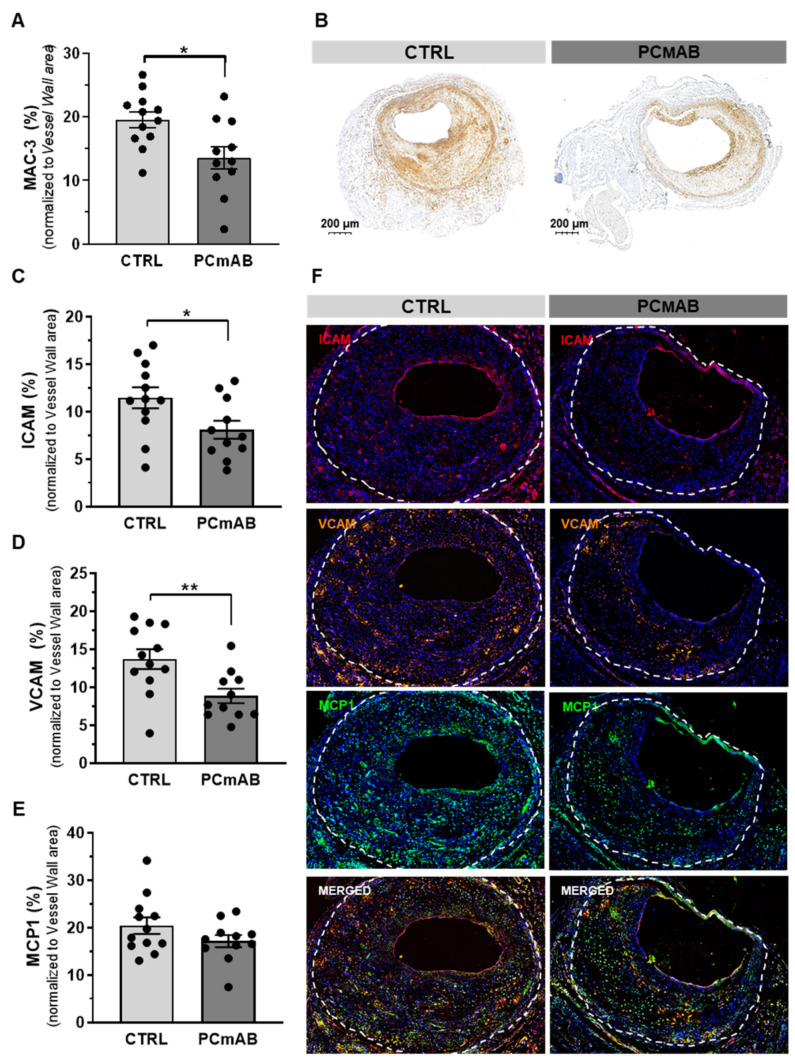
PC-mAb improves plaque inflammation by decreasing macrophage content and VCAM and ICAM expression. Quantification of % MAC-3 (**A**), Representative VG sections (**B**) and adhesion and inflammation associated factors ICAM-1 (**C**), VCAM-1 (**D**) and MCP-1 (**E**) expression in the CTRL (n = 12) and PC-mAb group (n = 11). Representative VG sections (**F**) Data presented as mean ± SEM. * *p* ≤ 0.05, ** *p* ≤ 0.01 by *t*-test.

**Figure 4 ijms-23-13662-f004:**
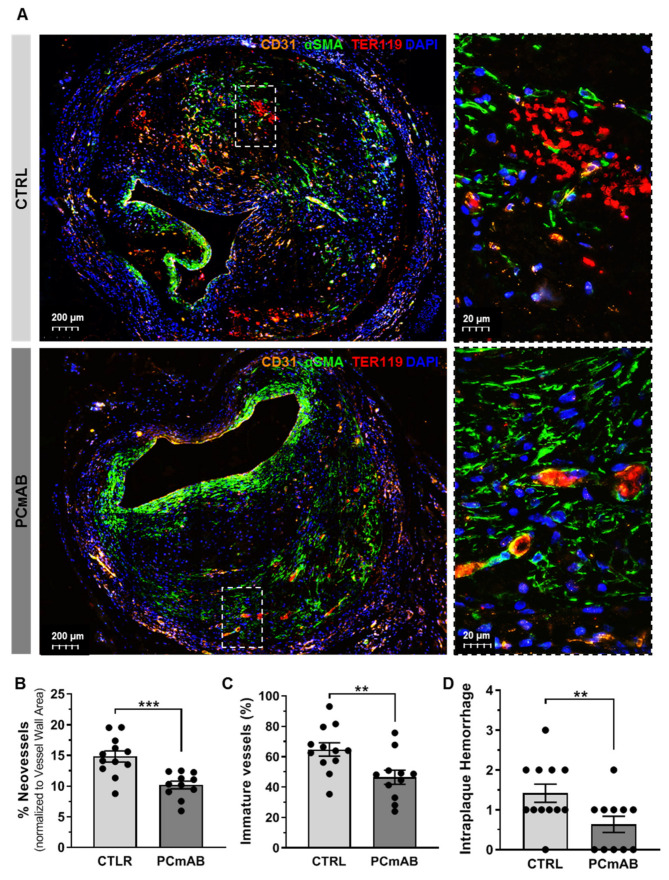
PC-mAb reduces plaque neovessel density and increases vessel maturity, reducing intraplaque hemorrhage. (**A**) Representative vein grafts cross sections of CD31 (orange), ACTA2 (green), TER119 (red) and DAPI (blue) staining of CTRL and PC-mAb group. Quantitative measurements of % Neovessels based on the CD31 staining (**B**) and % Immature Neovessels as shown by the lack of ACTA2 coverage (**C**) and Intraplaque Hemorrhage (Ter119+ erythrocytes outside the neovessel) (**D**) scoring in the CTRL (n = 12) and PC-mAb group (n = 11). Data presented as mean ± SEM. ** *p* ≤ 0.01, *** *p* ≤ 0.001 by *t*-test.

**Figure 5 ijms-23-13662-f005:**
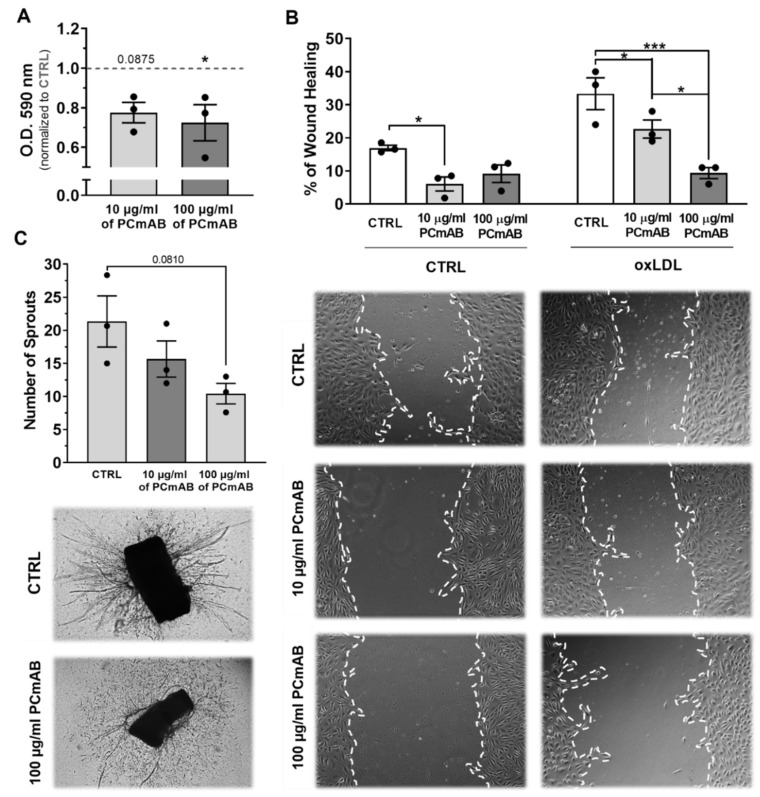
PC-mAb reduces HUVEC metabolic activity and HUVEC migration in vitro and neovessel sprouting ex vivo. Quantification of PC-mAb effects on the MMT assay (**A**), on the migration assay (**B**), and on the aortic ring assay (**C**). Representative images of the scratches on HUVEC mono-layers (**B**) treated with increasing doses of PC-mAb and with and without 5 µg/mL oxLDL, 16 h after scratching. Representative images of the aortic rings (**C**) treated with VEGF and PC-mAb. (**A**) Data normalized to CTRL group (indicated as 1 by a dashed line in the graph) and presented as mean ± SEM (n = 3). * *p* < 0.05, by 1-way ANOVA (*are significances versus control). (**B**) Data presented as mean ± SEM (n = 3). * *p* < 0.05, *** *p* < 0.001; by 2-way ANOVA. (**C**) Data presented as mean ± SEM (n = 3) by 1-way ANOVA.

**Figure 6 ijms-23-13662-f006:**
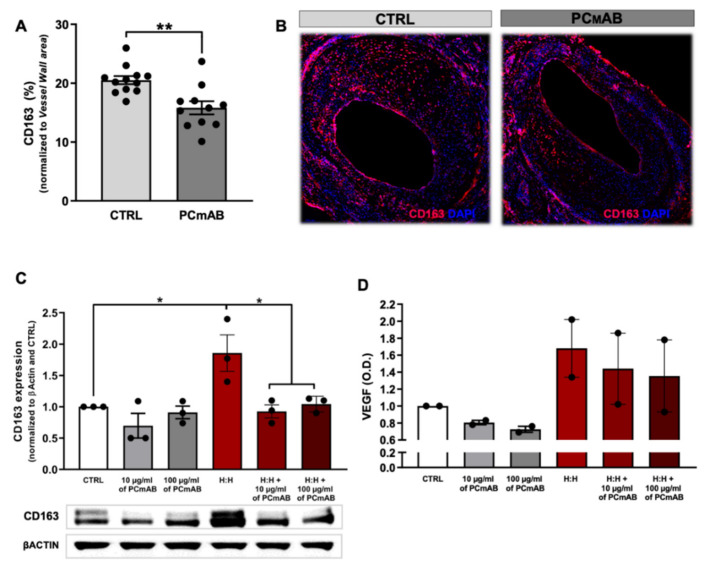
PC-mAb targets M(Hb) macrophages in vivo and in vitro by decreasing CD163 expression. Quantification of CD163 expression in VG lesions (**A**) and representative cross sections of CD163 (red) and DAPI (blue) staining of CTRL and PC-mAb group (**B**). Quantification of CD163 expression in THP-1 cells treated with increasing doses of PC-mAb and with and without Hb:Hp (H:H) (**C**).Quantification of VEFG levels in THP-1 cell supernatant (**D**). Data presented as mean ± SEM. * *p* ≤ 0.05, ** *p* ≤ 0.01 by *t*-test and by 1-way ANOVA.

## Data Availability

Data available upon reasonable request from corresponding author.

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
