# Peer review of "Phosphorylcholine Monoclonal Antibody Therapy Decreases Intraplaque Angiogenesis and Intraplaque Hemorrhage in Murine Vein Grafts"

_ijms, 2022, doi:10.3390/ijms232113662_

Round 1
Reviewer 1 Report
The authors demonstrate the therapeutic use of PC -mAB treatment in reducing intimal hyperplasia in murine vein grafts using a hypercholesterolemic male ApoE3*Leiden mice model.
The results shows that this treatment can reduce vessel wall thickening, however it had no effect on plaque features such as calcification. What does this say about the impact of PC -mAB treatment? Is it effective enough?
The model lacks endothelium as it’s known that endothelium is reduced/ lost in the early stages of IH development. The authors fail to discuss the importance of using HUVECs as an invitro model to show effectiveness of the mAb treatment.
The study looks at only limited markers at the protein level but show no to show any changes in RNA makers. Does PC-mAB have an impact on expression of other SMC markers as different markers are expressed at different stages leading to lesion development?
There are several errors in text including incomplete legends. ALSO, Figure 4D is missing.
Author Response
Reviewer 1
English language and style
( ) Extensive editing of English language and style required
(x) Moderate English changes required
( ) English language and style are fine/minor spell check required
( ) I don't feel qualified to judge about the English language and style
Yes Can be improved Must be improved Not applicable
Does the introduction provide sufficient background and include all relevant references?
( ) ( ) (x) ( )
Are all the cited references relevant to the research?
( ) ( ) (x) ( )
Is the research design appropriate?
( ) ( ) (x) ( )
Are the methods adequately described?
(x) ( ) ( ) ( )
Are the results clearly presented?
( ) ( ) (x) ( )
Are the conclusions supported by the results?
( ) ( ) (x) ( )
The authors demonstrate the therapeutic use of PC -mAB treatment in reducing intimal hyperplasia in murine vein grafts using a hypercholesterolemic male ApoE3*Leiden mice model.
The results shows that this treatment can reduce vessel wall thickening, however it had no effect on plaque features such as calcification. What does this say about the impact of PC -mAB treatment? Is it effective enough?
We thank the reviewer for the comment and based on our extensive data, we strongly believe that PCmAb treatment, is effective enough to attenuate vein graft disease. As shown in supplemental figure 2, there is no difference in plaque features (dissection, fibrin, foam cell, chondrocytes and calcification) when scored semi-quantitively in Masson Trichrome’s staining. In our morphometric analysis, we do see a significant reduction in intimal hyperplasia (22%) and a significant increase in lumen area (32%), which is crucial for vein graft patency. In addition, in our compositional analysis, we do find substantial evidence for improved plaque phenotype, demonstrated by increased collagen, reduced inflammatory markers (MAC3, ICAM, VCAM), improved IPA, diminished IPH. Overall, we therefore maintain our conclusion that PCmAb is able to reduce vein graft disease.
The model lacks endothelium as it’s known that endothelium is reduced/ lost in the early stages of IH development. The authors fail to discuss the importance of using HUVECs as an invitro model to show effectiveness of the mAb treatment.
We agree with the reviewer that it is important to mention the endothelium is damaged after the surgical procedure and that HUVECs are excellent to be used to study effectiveness of PC-mAb. We have added the following sentence to the discussion on page 12 line 297-299.
“Since it is well known that directly after vein graft procedure the endothelium is damaged [1] and activated we investigated the effects of PC-mAb on human venous endothelial cells (HUVECs).”
The study looks at only limited markers at the protein level but show no to show any changes in RNA makers. Does PC-mAB have an impact on expression of other SMC markers as different markers are expressed at different stages leading to lesion development?
We thank the reviewer for the comment. To investigate VSMC in the vein graft wall, we have decided to use ACTA2, which is generally accepted as a general VSMC marker. Indeed, we did not see a (significant) difference in ACTA2 content between both groups. Since there was no effect on ACTA2 content, and we did find a significant effect on MAC3, ICAM and VCAM content as well as IPA and IPH. Therefore, we did not further investigate the effect of other SMC markers but focused mainly on endothelial cells and macrophages.
There are several errors in text including incomplete legends. ALSO, Figure 4D is missing.
We apologies for the errors and the missing figure we went critically over the text and adapted the errors including the legends and added figure 4D.
Figure 4D is now added to the manuscript.
Page 8 line 194-195 Intraplaque Haemorrhage (Ter119+ erythrocytes outside the neovessel) (D)

Reviewer 2 Report
The article is devoted Ito the study of effects of human IgG1 monoclonal antibody against phosphorylcholine on vein graft disease. Evidences were obtained that this antibody attenuates intraplaque angiogenesis, improves neovessel maturation, reduces macrophage content in plaques, reduces ICAM and VCAM-expression and increases the content of collagen.
The paper is well written, the results obtained support the conclusions showing the potential of anti-phosphorylcholine antibodies for treatment of atherosclerosis (and vein graft disease) in humans.
However, there are certain minor issues which need to be addressed.
Introductory section would benefit from the addition of a little bit more information on what atherosclerosis is, what stages of atherosclerotic plaques development can be distinguished, what are the pathological factors contributing to atherosclerosis (the contribution of desialylated and oxidized LDL as well as the potential role of mutations of mitochondrial DNA should be mentioned). The role of oxLDL was discussed but it would be beneficial to add a little bit more information on receptor-mediated uptake and effects of oxidation specific epitopes of oxLDL. Some insight could be found for example here: DOI 10.3390/biomedicines9080915; DOI 10.3390/ijms22084080).
Page 3. Line 78. What is primary and secondary atherosclerosis?
References should follow each other in numerical order. On page 3 reference 31 goes after reference 27.
Page 4. Lines 120-121. There is no supplementary figure 3 in supplementary information.
Page 4. Lines 129-130. “However, the % MCP-1 did not vary between the two groups (Figure 3C and D)… ” The data on % MCP-1 are shown in Figure 3E.
Section D in Figure 4 is missing.
Figure 5 legend should mention the object studied.
Page 5. Line 217. Reference 8 should be put in square brackets.
Also references in the list of references should be in format required by mdpi rules.
Page 6. Line 235. Term oxPLs should be explained.
Page 6. Lines 247-249. “Mice were randomized based on their plasma cholesterol levels (Roche Diagnostics, kit 1489437) and body weight (supplemental figure 1). ” There is no such information in Supplemental figure 1.
The article can be accepted after minor revision.
Author Response
Reviewer 2
English language and style
( ) Extensive editing of English language and style required
( ) Moderate English changes required
(x) English language and style are fine/minor spell check required
( ) I don't feel qualified to judge about the English language and style
Yes Can be improved Must be improved Not applicable
Does the introduction provide sufficient background and include all relevant references?
( ) ( ) (x) ( )
Are all the cited references relevant to the research?
(x) ( ) ( ) ( )
Is the research design appropriate?
(x) ( ) ( ) ( )
Are the methods adequately described?
(x) ( ) ( ) ( )
Are the results clearly presented?
(x) ( ) ( ) ( )
Are the conclusions supported by the results?
(x) ( ) ( ) ( )
Comments and Suggestions for Authors
The article is devoted Ito the study of effects of human IgG1 monoclonal antibody against phosphorylcholine on vein graft disease. Evidences were obtained that this antibody attenuates intraplaque angiogenesis, improves neovessel maturation, reduces macrophage content in plaques, reduces ICAM and VCAM-expression and increases the content of collagen.
The paper is well written, the results obtained support the conclusions showing the potential of anti-phosphorylcholine antibodies for treatment of atherosclerosis (and vein graft disease) in humans.
We thank the reviewer for the comments and the effort that has been put in reviewing our manuscript.
However, there are certain minor issues which need to be addressed.
1) Introductory section would benefit from the addition of a little bit more information on what atherosclerosis is, what stages of atherosclerotic plaques development can be distinguished, what are the pathological factors contributing to atherosclerosis (the contribution of desialylated and oxidized LDL as well as the potential role of mutations of mitochondrial DNA should be mentioned). The role of oxLDL was discussed but it would be beneficial to add a little bit more information on receptor-mediated uptake and effects of oxidation specific epitopes of oxLDL. Some insight could be found for example here: DOI 10.3390/biomedicines9080915; DOI 10.3390/ijms22084080).
We thank the reviewer for the comment and have added extra background on atherosclerosis, oxLDL and mitochondrial DNA damage on page 3 line 99-108. We have added extra references (including the suggested article from the reviewever) as a result.
“Naïve atherosclerosis develops over decades, whereas accelerated atherosclerosis in vein grafts or in stents can be observed within months to years.[16] Apart from several dis-crepancies, the process of both native and accelerated atherosclerosis is relatively similar. Atherosclerosis is characterized by endothelial dysfunction and oxLDL-presence. Endo-thelial dysfunction leads to infiltration of leukocytes, predominantly macrophages, that aim to clear the oxLDL in the subendothelial layer. Upon oxLDL recognition and inter-nalization, macrophages undergo metabolic and functional reprogramming, ultimately resulting in foam cell formation.[17] In addition, oxLDL can results in mitochondrial DNA-damage in macrophages leading to cell death, demonstrating the crucial role of phospholipids in atherogenesis.[18]”
2) Page 3. Line 78. What is primary and secondary atherosclerosis?
We apologies for the unclear description. We have replaced this by the following sentence on page 3 line 97-99 “PC contributes via all these processes to native and accelerated atherosclerosis, the latter can occur after an intervention, such as stenting or bypass surgery.[1]”.
3) References should follow each other in numerical order. On page 3 reference 31 goes after reference 27.
We now updated the references and these are now in the proper numerical order.
4) Page 4. Lines 120-121. There is no supplementary figure 1 in supplementary information.
The supplemental figure 1 is now added on page 16 line 502-512
5) Page 4. Lines 129-130. “However, the % MCP-1 did not vary between the two groups (Figure 3C and D)… ” The data on % MCP-1 are shown in Figure 3E.
We corrected the sentence and it now reads as follows (page 5 line 169-170):” However, the % MCP-1 did not vary between the two groups (Figure 3E and F)”.
6) Section D in Figure 4 is missing.
Figure 4D is now added to the manuscript.
Page 8 line 194-195 Intraplaque Haemorrhage (Ter119+ erythrocytes outside the neovessel) (D)
7) Figure 5 legend should mention the object studied.
The legend of Figure 5 is adapted as described here.
Page 10 line 220-228 “Figure 5. PC-mAb reduces HUVEC metabolic activity and HUVEC migration in vitro and neovessel sprouting ex vivo. Quantification of PC-mAb effects on the MMT assay (A), on the migration assay (B) and on the aortic ring assay (C). Representative images of the scratches on HUVEC mono-layers (B) treated with increasing doses of PC-mAb and with and without 5 µg/ml oxLDL, 16 hours after scratching. Representative images of the aortic rings (C) treated with VEGF and PC-mAb. (A) Data normalized to CTRL group (indicated as 1 by a dashed red line in the graph) and presented as mean ± SEM (n=3). *P<0.05, **P<0.01; by 1-way ANOVA (* (in red) are significances versus control). (B) Data presented as mean ± SEM (n=3). *P<0.05, ***P<0.001; by 2-way ANOVA. (C) Data presented as mean ± SEM (n=3) by 1-way ANOVA.”
8) Page 5. Line 217. Reference 8 should be put in square brackets.
References are adapted and are now all annotated with square brackets including reference 8.
9) Also references in the list of references should be in format required by mdpi rules.
We updated the references and these are now in the proper numerical order and according the MDPI format.
10) Page 6. Line 235. Term oxPLs should be explained.
We have written the term oxidized phospholipids throughout the manuscript without abbreviation.
11) Page 6. Lines 247-249. “Mice were randomized based on their plasma cholesterol levels (Roche Diagnostics, kit 1489437) and body weight (supplemental figure 1). ” There is no such information in Supplemental figure 1.
We regret that supplemental figure 1 was not in the previous version of the manuscript this supplemental figure is now added on page 16 line 502-512 (see for figure Q4) .
The article can be accepted after minor revision.

Round 2
Reviewer 1 Report
The authors have answered all concerns and addressed the changes.